# Four-Term Finite-Sample OOD Transfer Bound for Fourier Neural Operators

**Sebastien Kawada**

Kaons

Los Angeles, CA, USA

sebastien@kaons.org

## Abstract

Fourier Neural Operators are increasingly deployed on distributions and meshes that differ from training, yet no unified framework decomposes the resulting transfer error into its distinct sources. We prove a high-probability finite-sample excess-risk bound for nonlinear FNOs on periodic domains that separates transfer error into four interpretable channels: statistical complexity, Sobolev discretization, Wasserstein distribution shift, and train-mesh/test-mesh aliasing. The aliasing channel, which formalizes how nonlinear processing on mismatched grids folds unresolved frequencies into retained modes, is the distinctive new component. Term-wise diagnostics on Darcy, Helmholtz, and Burgers problems confirm the predicted sign and scaling of each channel. A factorial experiment disentangles the effects of network depth and spectral normalization, and constraining operator norms substantially reduces the bound-to-excess gap. The decomposition identifies which degradation channel dominates in a given transfer scenario, providing actionable diagnostic guidance beyond what a single aggregate bound can offer.

## 1 Introduction

Neural operators such as Fourier Neural Operators (FNOs) are now standard surrogates for PDE families Li et al. (2021); Kovachki et al. (2023). Other influential architectures include DeepONet Lu et al. (2021). In deployment, however, train and test conditions rarely match. Models are fit on finite samples from one distribution and one mesh, then evaluated on different distributions and resolutions. This train–test mismatch introduces multiple degradation sources that are often studied separately. We focus on the regime where sample size, distribution, and mesh all shift simultaneously, and we target one finite-sample excess-risk inequality that isolates the dominant channels in a single bound.

Our bound decomposes target excess risk into four channels (Figure 1):

   (i) statistical complexity with concentration,
  (ii) Sobolev discretization,
 (iii) Wasserstein distribution shift, and
 (iv) train-mesh/test-mesh aliasing mismatch.

Recent theory addresses these components in isolation. Prior work covers operator approximation and universality Kovachki et al. (2023); Lanthaler et al. (2025); Marcati & Schwab (2023), discretization and mesh effects Lanthaler et al. (2024); Gao et al. (2025); Furuya et al. (2024), statistical complexity Kim & Kang (2024), OOD transfer Lara Benitez et al. (2024), and global theory–practice gaps Grohs et al. (2025). We contribute a single theorem that combines them for nonlinear FNOs on periodic domains.

The aliasing component is the distinctive coupling in this decomposition. Even with fixed mode truncation, nonlinear processing on different grids can fold unresolved frequencies into retained low modes in different ways. We formalize this via an explicit mismatch functional and pair the theorem with Fourier-space diagnostics that visualize the lattice-collision mechanism.

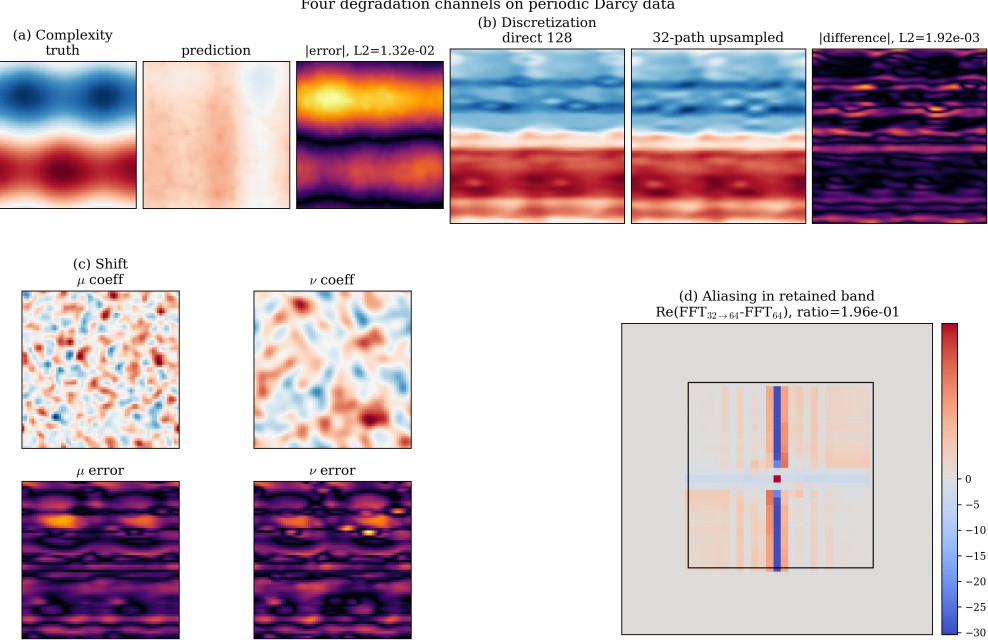

Figure 1: Four degradation channels on periodic Darcy data: (a) finite-sample prediction error, (b) coarse-vs-fine path discretization gap, (c) source/target input shift and induced error change, and (d) retained-band spectral contamination under mesh mismatch.

**Contributions.**

(C1) A high-probability finite-sample transfer theorem for nonlinear FNOs that separates complexity/concentration, discretization, Wasserstein shift, and mesh-mismatch aliasing in one inequality.

(C2) A complete proof under one explicit Sobolev assumption set, with explicit constants for each addend.

(C3) Term-wise diagnostics on Darcy and Helmholtz, plus isolated Burgers evidence and Fourier-space visualizations that directly expose aliasing and mesh-transfer effects.

## 2  RELATED WORK AND GAP POSITIONING

Table 1 summarizes the closest lines of work and identifies which degradation channels each addresses.

Prior work addresses these degradation channels separately. Transfer bounds Lara Benitez et al. (2024) omit mesh effects. Discretization analyses Lanthaler et al. (2024); Gao et al. (2025) assume a fixed data distribution. Complexity bounds Kim & Kang (2024); Grohs et al. (2025) do not account for mesh mismatch. Our theorem is the first to contain all four channels for nonlinear FNOs in a single finite-sample inequality, with the explicit aliasing recursion through nonlinear layers as the main new ingredient.

Our shift term connects to classical domain-adaptation decompositions Ben-David et al. (2007); Mansour et al. (2009); Ben-David et al. (2010), with Wasserstein distance playing the discrepancy role in our Sobolev-function setting. Convergence-rate analysis for learning linear operators from noisy data de Hoop et al. (2023) addresses statistical convergence but without OOD transfer. Grohs et al. Grohs et al. (2025) quantify the gap between theoretical approximation rates and practical training performance, but their analysis does not decompose OOD transfer into separate channels. Marcati and Schwab Marcati & Schwab (2023) prove exponential convergence of DeepONet in the

Table 1: Feature matrix: degradation channels covered by nearby neural-operator theory (2022–2026). ✓ = explicitly addressed; — = absent.

| Reference | Fin.-samp. | Complex. | Discret. | Shift | Aliasing | Nonlin. FNO |
|---|---|---|---|---|---|---|
| Kovachki et al. Kovachki et al. (2023) | — | ✓ | ✓ | — | — | ✓ |
| Lanthaler et al. Lanthaler et al. (2025) | — | — | — | — | — | ✓ |
| Lingsch et al. Lingsch et al. (2024) | — | — | ✓ | — | — | ✓ |
| Lara Benitez et al. Lara Benitez et al. (2024) | ✓ | ✓ | — | ✓ | — | — |
| Kim & Kang Kim & Kang (2024) | ✓ | ✓ | — | — | — | ✓ |
| Lanthaler et al. Lanthaler et al. (2024) | — | — | ✓ | — | — | ✓ |
| Subedi & Tewari Subedi & Tewari (2025) | ✓ | ✓ | ✓ | — | — | — |
| Gao et al. Gao et al. (2025) | — | — | ✓ | — | ✓ | ✓ |
| Furuya et al. Furuya et al. (2024) | — | — | ✓ | — | — | ✓ |
| Grohs et al. Grohs et al. (2025) | ✓ | ✓ | — | — | — | ✓ |
| **This work** | ✓ | ✓ | ✓ | ✓ | ✓ | ✓ |

approximation-theoretic (infinite-sample, fixed-distribution) regime, complementary to our setting, which includes finite samples and distribution shift.

## 3 SETUP

Let $\mathbb{T}^2 = [0,1]^2$ denote the periodic torus. Inputs and outputs belong to Sobolev spaces

$$\mathcal{X} = H^s(\mathbb{T}^2; \mathbb{R}^{d_a}), \qquad \mathcal{Y} = H^s(\mathbb{T}^2; \mathbb{R}^{d_u}), \qquad s > 1.$$

Let $G^\star : \mathcal{X} \to \mathcal{Y}$ be $H^s \to H^s$ Lipschitz.

We consider an FNO hypothesis class $\mathcal{G}_{K,L,W}$ with depth $L$, width $W$, Fourier truncation $K$, smooth activation (GELU or tanh), and bounded operator norms. We sample training data from $\mu$ on an $N \times N$ grid ($h = 1/N$) and measure test risk under $\nu$ on an $N' \times N'$ grid ($h' = 1/N'$).

For a loss $\ell : \mathcal{Y} \times \mathcal{Y} \to \mathbb{R}_+$ and a predictor $G$, define

$$\mathcal{E}_{\rho,h}(G) := \mathbb{E}_{(a,u)\sim\rho}\left[\ell(G_h(a), u_h)\right], \qquad \widehat{\mathcal{E}}_{\mu,h}(G) := \frac{1}{n}\sum_{i=1}^n \ell(G_h(a_i), u_{i,h}).$$

We use the product metric on sample space

$$d\big((a,u),(a',u')\big) := \|a - a'\|_{L^2} + \|u - u'\|_{L^2},$$

and $W_{1,d}(\mu,\nu)$ denotes Wasserstein-1 induced by $d$. On bounded outputs, we assume $\ell$ is jointly Lipschitz:

$$|\ell(\hat{u}_1, u_1) - \ell(\hat{u}_2, u_2)| \le L_\ell(\|\hat{u}_1 - \hat{u}_2\|_{L^2} + \|u_1 - u_2\|_{L^2}),$$

and $|\ell| \le R_\ell$ on this regime.

For the generalization term, we use the mesh-$h$ Rademacher complexity of the model class,

$$\mathfrak{R}_n(\mathcal{G}_{K,L,W}; h) := \mathbb{E}_\varepsilon\left[\sup_{G\in\mathcal{G}_{K,L,W}} \frac{1}{n}\left\|\sum_{i=1}^n \varepsilon_i\, G_h(a_i)\right\|_{L^2}\right],$$

where $\varepsilon_i$ are i.i.d. Rademacher random variables and $n$ is the training sample size. Since $G \in \mathcal{G}_{K,L,W}$ uses $K$-mode Fourier truncation, we evaluate risks on reconstructed outputs; inserting $\Pi_{\le K}$ before $\ell$ is equivalent under this realization convention.

Different levels of averaging are needed because the raw mismatch $\Delta_G$ depends on both the model and the specific input, while the theorem requires a bound that holds uniformly. To separate mesh mismatch effects, we use four related aliasing quantities. The per-input per-model quantity $\Delta_G$ is the finest-grained. Averaging over inputs gives $\mathcal{A}_{\rho_a}$. The distribution-aware envelope $\overline{\mathcal{A}}_{\mu,\nu}$ appears

in the theorem. The worst-case $\mathcal{A}_{\mathrm{sup}}$ is distribution-agnostic. Let $D_N$ be periodic sampling on an $N \times N$ grid, $D_N^{-1}$ the trigonometric reconstruction operator, and

$$\mathcal{I}_{h' \leftarrow h} := D_{N'} D_N^{-1}$$

the mesh-transfer interpolation map from mesh $h$ to mesh $h'$.

**Mesh-consistency of input lifting.** We require $\Delta_0 = 0$ in the aliasing recursion (Appendix E), meaning the input projection is mesh-consistent in the retained Fourier band. This holds when $\mathcal{P}_{\mathrm{in}}$ acts on the low-mode coefficients shared by both grids. Define

$$\Delta_G(a; K, h, h') := \|\Pi_{\leq K} \mathcal{I}_{h' \leftarrow h} G_h(a) - \Pi_{\leq K} G_{h'}(\mathcal{I}_{h' \leftarrow h} a)\|_{L^2}.$$

For an input distribution $\rho_a$ on $\mathcal{X}$,

$$\mathcal{A}_{\rho_a}(G; K, h, h') := \mathbb{E}_{a \sim \rho_a}[\Delta_G(a; K, h, h')].$$

We use

$$\overline{\mathcal{A}}_{\mu,\nu}(K, h, h') := \sup_{G \in \mathcal{G}_{K,L,W}} (\mathcal{A}_{\mu_a}(G; K, h, h') + \mathcal{A}_{\nu_a}(G; K, h, h'))$$

as the distribution-aware mesh-mismatch term, and

$$\mathcal{A}_{\mathrm{sup}}(K, h, h') := \sup_{G \in \mathcal{G}_{K,L,W}} \sup_{\|a\|_{H^s} \leq B_x} \Delta_G(a; K, h, h')$$

as the worst-case envelope. Here $\Pi_{\leq K}$ is low-mode projection. The four quantities form a nested hierarchy, from the finest-grained per-input mismatch $\Delta_G$ through distribution-averaged and distribution-aware envelopes to the worst-case $\mathcal{A}_{\mathrm{sup}}$. The theorem uses $\overline{\mathcal{A}}_{\mu,\nu}$.

**Notation summary.**

| Symbol | Meaning |
|---|---|
| $G^\star, \mathcal{G}_{K,L,W}$ | True operator; FNO hypothesis class (modes $K$, depth $L$, width $W$) |
| $\mu, \nu$ | Train / test data distributions |
| $h = 1/N, h' = 1/N'$ | Train / test mesh spacings |
| $\mathfrak{R}_n$ | Model-class Rademacher complexity on mesh $h$ |
| $W_{1,d}(\mu, \nu)$ | Wasserstein-1 distance on $\mathcal{X} \times \mathcal{Y}$ |
| $\overline{\mathcal{A}}_{\mu,\nu}$ | Distribution-aware aliasing envelope (appears in Theorem 1) |
| (S1)–(S4) | Assumption items below |

**Assumption 1** (Boundedness and smoothness). *There exist constants $B_x, B_y, M_\theta > 0$ such that:*

(S1) *$\|a\|_{H^s} \leq B_x$ and $\|u\|_{H^s} \leq B_y$ almost surely under both $\mu$ and $\nu$.*

(S2) *All spectral and pointwise FNO layer operators have $L^2 \to L^2$ norm at most $M_\theta$.*

(S3) *Activations are $C^2$ with $\|\sigma'\|_\infty \leq B_\sigma$ and $\|\sigma''\|_\infty \leq B'_\sigma$.*

(S4) *(Hidden-state Sobolev propagation) For every $G \in \mathcal{G}_{K,L,W}$ and every input in $\mathrm{supp}(\mu_a) \cup \mathrm{supp}(\nu_a)$, all layer states satisfy $\|v_t\|_{H^s} \leq C_{\mathrm{hid}}$ for $t = 0, \ldots, L$ on both meshes $N, N'$.*

Assumption 1(S4) is a uniform regularity envelope (class-level, both meshes, both supports). It is the only non-standard step used to propagate Sobolev control through nonlinear depth in our discretization and aliasing bounds. Proposition 1 in Appendix F derives (S4) for $s=2$, $d=2$ with $C^2$ activations. A quadratic recursion $R_{t+1} = \alpha_t R_t + \beta R_t^2$ determines an explicit $C_{\mathrm{hid}}$ from $(B_x, M_\theta, B_\sigma, \|\mathcal{P}_{\mathrm{in}}\|, L)$. Under spectral norm constraints making $\alpha_t < 1$ for all layers, the recursion contracts and $C_{\mathrm{hid}} = O(\|\mathcal{P}_{\mathrm{in}}\| B_x)$. In general, $C_{\mathrm{hid}}$ can grow exponentially in $L$. This is the same mechanism underlying the global Lipschitz envelope.

## 4 MAIN THEOREM

Let

$$\widehat{\mathcal{E}}_{\mu,h}(\hat{G}) \leq \inf_{G \in \mathcal{G}_{K,L,W}} \widehat{\mathcal{E}}_{\mu,h}(G) + \varepsilon_{\mathrm{erm}}$$

define an $\varepsilon_{\mathrm{erm}}$-approximate empirical risk minimizer on training mesh $h$.

**Theorem 1** (Four-term finite-sample OOD transfer bound). *Assume:*

*(A1) periodic domain $\mathbb{T}^2$,*

*(A2) smooth activation (GELU/tanh),*

*(A3) uniform Cartesian grids,*

*(A4) mode truncation $K < \min(N, N')/2$,*

*(A5) Assumption 1.*

*Then for any $\delta \in (0,1)$, with probability at least $1 - \delta$,*

$$\mathcal{E}_{\nu,h'}(\hat{G}) - \inf_{G \in \mathcal{G}_{K,L,W}} \mathcal{E}_{\nu,h'}(G) \leq C_1\, \mathfrak{R}_n(\mathcal{G}_{K,L,W}; h) + C_2\, (h^s + h'^s) + C_3\, W_{1,d}(\mu, \nu)$$

$$+ C_4\, \overline{\mathcal{A}}_{\mu,\nu}(K, h, h') + C_5 \sqrt{\frac{\log(1/\delta)}{n}} + \varepsilon_{\mathrm{erm}}.$$

*One explicit choice is*

$$C_1 = 4L_\ell, \quad C_2 = 2L_\ell C_{\mathrm{disc}}, \quad C_3 = 2L_\ell(1 + L_G), \quad C_4 = 2L_\ell, \quad C_5 = 4R_\ell,$$

*where $L_G$ is a global Lipschitz envelope for $G \in \mathcal{G}_{K,L,W}$ induced by $(M_\theta, B_\sigma, L)$, $C_{\mathrm{disc}}$ is the Sobolev discretization constant, and $R_\ell$ bounds $|\ell|$ on the bounded output regime.*

The six addends group into four degradation channels: (i) statistical complexity ($C_1\mathfrak{R}_n + C_5\sqrt{\cdot}$, both arising from the generalization step in Lemma 1), (ii) discretization ($C_2(h^s + h'^s)$), (iii) distribution shift ($C_3 W_{1,d}$), and (iv) aliasing ($C_4\overline{\mathcal{A}}_{\mu,\nu}$). The ERM slack $\varepsilon_{\mathrm{erm}}$ vanishes for exact minimizers and is not counted as a channel. The $h'^s$ contribution to the discretization term arises because the proof bridges train and test meshes through the continuum for both $\hat{G}$ and the comparator; when $h' \ll h$, this term is dominated by $h^s$. The additive structure follows from triangle inequalities and uniform envelopes. It should not be read as statistical independence of channels; interaction effects are absorbed into the concrete prefactors and envelopes.

**Remark 1.** *If $\varepsilon_{\mathrm{erm}} = 0$, the statement reduces to the exact-ERM case.*

**Remark 2.** *No assumptions beyond (A1)–(A5) and the loss/metric definitions from Setup are introduced during the proof reduction. Under Assumption 1(S1), $\overline{\mathcal{A}}_{\mu,\nu}(K, h, h') \leq 2\mathcal{A}_{\mathrm{sup}}(K, h, h')$, yielding a distribution-agnostic corollary.*

An explicit rate for $\mathfrak{R}_n$ follows from Kim and Kang Kim & Kang (2024); see Appendix B.

**Remark 3** (Choice of $W_1$ for distribution shift). *We use Wasserstein-1 because Kantorovich–Rubinstein duality pairs directly with Lipschitz loss composition, yielding the clean constant $C_3 = 2L_\ell(1 + L_G)$. Discrepancy distance Ben-David et al. (2010); Cortes & Mohri (2011) or Rényi divergence would require estimating suprema over the hypothesis class, introducing additional computational challenges without improving the bound's structural decomposition.*

**Corollary 1** (Per-layer constants). *If per-layer operator norms $\|\mathcal{K}_t\|$, $\|W_t\|$ are available, the geometric sum in $C_{\mathrm{disc}}$ and $C_{\mathrm{alias}}$ becomes $\sum_{j=0}^{L-1} \prod_{i=j+1}^{L-1} B_\sigma(\|\mathcal{K}_i\| + \|W_i\|)$, replacing the uniform geometric sum $\sum_{j=0}^{L-1} (2B_\sigma M_\theta)^j$. In the depth-ablation experiments, per-layer constants reduce the bound-to-excess gap, with the largest improvements (up to three orders of magnitude) occurring at greater depths where the uniform envelope is most conservative.*

The decomposition also permits modular replacement of individual channel bounds. As a concrete example, we can swap the Rademacher complexity channel for a PAC-Bayes bound while leaving the other three channels unchanged.

**Corollary 2** (PAC-Bayes complexity variant)**.** *Let $P = \mathcal{N}(w_0, \lambda^2 I)$ be a prior on the parameter space of $\mathcal{G}_{K,L,W}$ chosen before seeing data, and let $Q = \mathcal{N}(\hat{w}, \lambda^2 I)$ be the posterior centered at the trained weights $\hat{w}$. Then the bound of Theorem 1 holds with $\mathcal{E}_{\nu,h'}(\hat{G})$ replaced by $\mathbb{E}_{G\sim Q}[\mathcal{E}_{\nu,h'}(G)]$ on the left-hand side, and the complexity and concentration terms $C_1\mathfrak{R}_n + C_5\sqrt{\cdot}$ replaced by the PAC-Bayes bound*

$$\sqrt{\frac{\mathrm{KL}(Q\|P) + \log(2\sqrt{n}/\delta)}{2n}}, \qquad \mathrm{KL}(Q\|P) = \frac{\|\hat{w} - w_0\|^2}{2\lambda^2},$$

*while the discretization, shift, and aliasing terms remain unchanged. The proof (Appendix G) replaces Lemma 1 with the McAllester bound McAllester (1999); the other two lemmas are unaffected.*

## 5 PROOF ROADMAP AND MAIN REDUCTION

Let

$$\bar{G} \in \mathcal{G}_{K,L,W}$$

be an arbitrary comparator. The natural bridge between source risk on mesh $h$ and target risk on mesh $h'$ is the $\mu$-risk evaluated on both meshes, which separates the generalization, shift, and mesh-transfer contributions. For $\hat{G}$, insert and subtract $\mu$-risk terms at meshes $h, h'$:

$$\mathcal{E}_{\nu,h'}(\hat{G}) - \mathcal{E}_{\nu,h'}(\bar{G}) \le \underbrace{\mathcal{E}_{\mu,h}(\hat{G}) - \mathcal{E}_{\mu,h}(\bar{G})}_{\Gamma_{\mathrm{gen}}}$$
$$+ \underbrace{\left|\mathcal{E}_{\nu,h'}(\hat{G}) - \mathcal{E}_{\mu,h'}(\hat{G})\right| + \left|\mathcal{E}_{\mu,h'}(\bar{G}) - \mathcal{E}_{\nu,h'}(\bar{G})\right|}_{\Gamma_{\mathrm{shift}}}$$
$$+ \underbrace{\left|\mathcal{E}_{\mu,h'}(\hat{G}) - \mathcal{E}_{\mu,h}(\hat{G})\right| + \left|\mathcal{E}_{\mu,h'}(\bar{G}) - \mathcal{E}_{\mu,h}(\bar{G})\right|}_{\Gamma_{\mathrm{mesh}}}. \qquad (1)$$

This is a model-pair decomposition in $(\hat{G}, \bar{G})$.

**ERM reduction.** By $\varepsilon_{\mathrm{erm}}$-ERM optimality of $\hat{G}$,

$$\Gamma_{\mathrm{gen}} \le 2 \sup_{G \in \mathcal{G}_{K,L,W}} \left|\widehat{\mathcal{E}}_{\mu,h}(G) - \mathcal{E}_{\mu,h}(G)\right| + \varepsilon_{\mathrm{erm}}.$$

**Lemma 1** (Complexity envelope)**.** *Under Assumption 1, with probability at least $1 - \delta$,*

$$\Gamma_{\mathrm{gen}} \le 4L_\ell \,\mathfrak{R}_n(\mathcal{G}_{K,L,W}; h) + 4R_\ell \sqrt{\frac{\log(1/\delta)}{n}}.$$

*Sketch.* Apply symmetrization and contraction for Rademacher complexity Bartlett & Mendelson (2002); Shalev-Shwartz & Ben-David (2014), then bounded-difference concentration Boucheron et al. (2013).

**Lemma 2** (Shift envelope)**.** *If $G \mapsto \ell(G_h(\cdot), \cdot)$ is $L_\ell(1 + L_G)$-Lipschitz on $(\mathcal{X} \times \mathcal{Y}, d)$, then*

$$\Gamma_{\mathrm{shift}} \le 2L_\ell(1 + L_G)\, W_{1,d}(\mu, \nu).$$

*Sketch.* Apply Kantorovich–Rubinstein duality to each shift term in (1) under metric $d$ Villani (2009); Peyré & Cuturi (2019).

**Lemma 3** (Mesh transfer = discretization + aliasing)**.** *For any $G \in \mathcal{G}_{K,L,W}$,*

$$|\mathcal{E}_{\mu,h'}(G) - \mathcal{E}_{\mu,h}(G)| \le L_\ell C_{\mathrm{disc}}(h^s + h'^s) + L_\ell \mathcal{A}_{\mu_a}(G; K, h, h').$$

*Consequently,*

$$\Gamma_{\mathrm{mesh}} \le 2L_\ell C_{\mathrm{disc}}(h^s + h'^s) + 2L_\ell \overline{\mathcal{A}}_{\mu,\nu}(K, h, h').$$

*Sketch.* Because the loss couples model output with labels on a specific mesh, risks at different resolutions cannot be compared directly. Adding and subtracting continuum risk decouples the mesh from the model, isolating per-mesh Sobolev interpolation error and residual mesh mismatch. The model-specific aliasing is then replaced by the distribution-aware envelope (Figure 2).

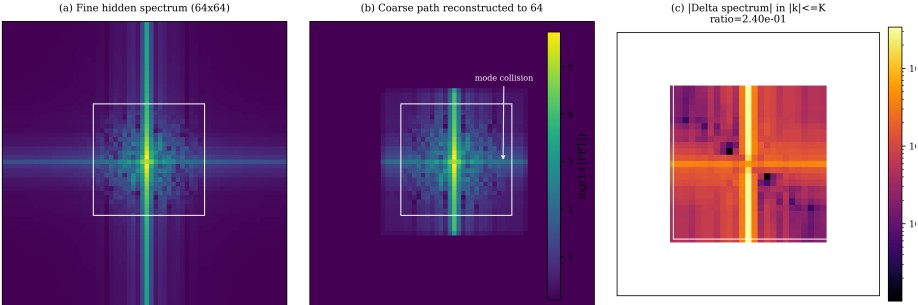

Figure 2: Aliasing anatomy in Fourier space. High-frequency content from coarse-grid processing folds into retained low modes, matching the lattice-collision mechanism used in the aliasing lemma.

**Combination.** Substituting the three lemmas into (1) and taking suprema over the hypothesis class yields Theorem 1 with $C_1, \ldots, C_5$. Because the bound is uniform in $\bar{G}$, the infimum over $\bar{G} \in \mathcal{G}_{K,L,W}$ recovers the excess-risk form.

## 6 EXPERIMENTAL SETUP AND DIAGNOSTICS

The experiments below aim to validate the sign structure and scaling directions predicted by Theorem 1, not to recover tight constants. We evaluate on Darcy (elliptic/smooth) and Helmholtz (oscillatory) problems, following common operator-learning benchmark conventions Li et al. (2021); Takamoto et al. (2022), with additional isolated Burgers evidence summarized in Table 6 (Extended Results).

**Shifts.** For Darcy, we induce distribution shift by varying the covariance length scale. For Helmholtz, we vary both the length scale and the forcing frequency.

**Meshes and modes.** Train meshes use $N \in \{32, 64\}$ in the base sweep and $N \in \{16, 24, 32, 48, 64, 96\}$ in the expanded scaling sweep. Test meshes use $N' \in \{48, 64, 96, 128\}$, and mode truncations satisfy $K < \min(N, N')/2$ in all runs.

**Diagnostics.** We estimate empirical proxies for each theorem addend and compare the resulting bound right-hand side with two excess-risk proxies: (i) variance-baseline excess and (ii) seed-best excess. We report both as diagnostics and do not interpret either as a tight estimator of the theorem's excess-risk quantity.

**Shift proxy and computability.** The theorem uses $W_{1,d}(\mu, \nu)$ on $\mathcal{X} \times \mathcal{Y}$. In experiments we use a projected surrogate

$$\widetilde{W}_1 := W_1((P_m)_{\#}\mu_a, (P_m)_{\#}\nu_a),$$

where $P_m$ is a PCA projection fitted on training inputs. This proxy is used only for diagnostics and trend comparisons. When labels are generated by a deterministic Lipschitz map $u = G^{\star}(a)$, one has $W_{1,d}(\mu, \nu) \leq (1 + \mathrm{Lip}(G^{\star})) W_1(\mu_a, \nu_a)$ under metric $d$, so projected input-space transport can still track shift trends up to a constant distortion.

**Sanity checks before sweeps.** Before large runs, we verify:

(V1) aliasing is numerically near-zero for identical meshes on identity/same-grid polynomial maps,

(V2) aliasing increases under train/test mesh mismatch for nonlinear maps,

(V3) discretization decreases with finer training resolution,

(V4) shift proxies increase under controlled perturbation magnitude.

Table 2: Experiment 1 (Darcy-medium): unconstrained vs spectrally constrained FNO. Tight constants use per-layer operator norms; uniform constants use the class-level envelope $M_\theta$.

| Model | $L$ | $M_\theta$ | $L_G$ (tight) | Bound RHS (tight) | Excess (seed-best) | $\log_{10}$ gap (tight) | $\log_{10}$ gap (uniform) |
|---|---|---|---|---|---|---|---|
| unconstrained | 4 | 6.006 | 6.731e+03 | 3.613e+03 | 6.953e-06 | 8.72 | 9.59 |
| spectral_norm_0.5 | 2 | 0.500 | 2.227e+00 | 1.366e+00 | 6.038e-07 | 6.35 | 6.37 |

Table 3: Factorial design: depth $\times$ spectral normalization (Darcy, 10 seeds per cell). Disentangling depth and SN shows both independently reduce the bound-to-excess gap.

| $L$ | SN | $M_\theta$ | $L_G$ | Bound | Excess | $\log_{10}$ gap |
|---|---|---|---|---|---|---|
| 2 | No | 4.465 | 7.333e+01 | 2.193e+01 | 4.281e-06 | 6.75 |
| 2 | Yes | 0.504 | 2.629e+00 | 1.349e+00 | 6.551e-07 | 6.31 |
| 4 | No | 5.702 | 1.070e+04 | 9.460e+03 | 9.123e-06 | 8.95 |
| 4 | Yes | 0.502 | 4.425e+00 | 4.352e+00 | 7.006e-07 | 6.79 |

## 7 EMPIRICAL RESULTS

We now report term-wise diagnostics and compare the resulting bound with two excess-risk proxies (variance-baseline and seed-best excess).

**Constrained vs. unconstrained (Table 2).** Experiment 1 shows that constraining layer operator norms and reducing depth reduces the seed-best $\log_{10}$ gap from 8.72 to 6.35.

**Factorial design (Table 3).** The factorial design disentangles the separate contributions of depth and spectral normalization. Seed-variance analysis (Table 4) confirms stability across random initializations.

**Scaling sweeps (Figure 3).** In the expanded scaling sweep, complexity and concentration terms decrease with $n$, while discretization and aliasing proxies increase with $h$. Both trends match predicted signs. Under controlled shift interpolation, observed excess increases monotonically with projected transport distance, consistent with the linear envelope predicted by the $C_3 W_{1,d}$ term.

**Depth ablation (Figure 4).** Depth-ablation diagnostics expose the exponential worst-case-envelope effect and show substantial reduction when per-layer constants are used (Corollary 1). We were surprised by how much the per-layer tracking helped at depth 6–8, where the uniform envelope is orders of magnitude more conservative than the product of measured layer norms.

**Practical interpretation of the four channels.** The empirical evidence supports using Theorem 1 as a qualitative trend predictor rather than a numerically tight bound. Across all tested regimes, the sign structure is stable. Reducing sample size increases the statistical channel. Coarsening training meshes increases discretization and aliasing channels. Stronger source-to-target perturbations increase transport-based shift proxies. This sign stability is useful in practice because it identifies which parameter to modify when transfer quality degrades.

The experiments suggest three practical guidelines. First, constraining layer operator norms and limiting depth lowers the dominant Lipschitz envelopes and substantially reduces observed bound-to-excess gaps (Table 2). Second, mode budgets should remain in moderate $K/N$ regimes for mesh transfer, since aliasing envelopes deteriorate as $K$ approaches Nyquist (Appendix E). Third, projected Wasserstein diagnostics should be read as monotone shift indicators, not plug-in estimates of $W_{1,d}(\mu, \nu)$, because projection compresses high-dimensional transport geometry.

Extended sweep summaries, Burgers evidence, and a mesh-transfer gallery appear in the appendix.

**Prospective channel prediction.** To test whether the decomposition provides actionable guidance, we design two held-out settings and predict the dominant channel *before* evaluation. In Setting A (mesh-dominated: $N{=}32$, $N'{=}128$, same distribution), we predict discretization and aliasing will

Table 4: Seed-variance summary over 10 seeds per primary configuration.

| pde | size | n_runs | excess (seed-best) mean±std | $\log_{10}$ gap mean±std |
|---|---|---|---|---|
| darcy | medium | 10 | 1.635e-05 ± 7.802e-06 | 8.48 ± 1.98 |
| darcy | small | 10 | 9.134e-06 ± 8.610e-06 | 8.26 ± 0.78 |
| helmholtz | medium | 10 | 1.125e-06 ± 1.690e-07 | 6.71 ± 0.51 |
| helmholtz | small | 10 | 2.727e-06 ± 1.830e-06 | 7.00 ± 0.33 |

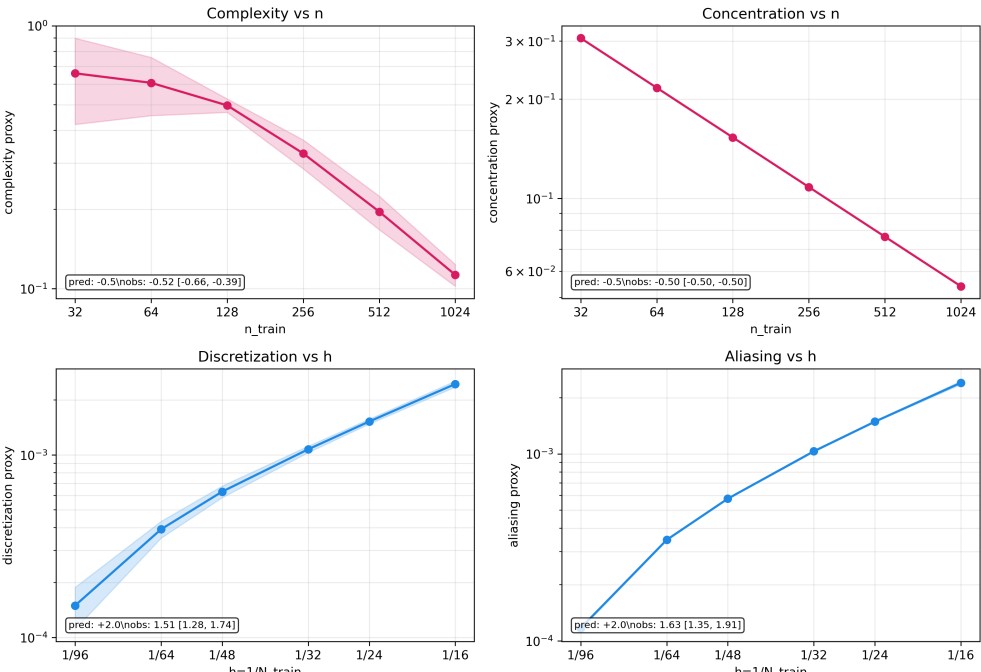

Figure 3: Expanded scaling sweeps (six points per axis, five seeds each) with observed slopes and 95% confidence intervals.

dominate. In Setting B (shift-dominated: $N=N'=64$, large length-scale shift), we predict the shift channel will dominate.

Setting B is correctly predicted: shift is the largest term ($\widetilde{W}_1 \approx 1.16$ versus complexity $\approx 0.35$ and near-zero discretization/aliasing), and the top-two ordering matches. Setting A is not: the projected $\widetilde{W}_1$ proxy reports a shift of $\approx 1.47$ even when train and test draws come from the same GRF, because finite-sample variation induces nonzero empirical transport distance. This was the most informative failure in our experiments. It shows that $\widetilde{W}_1$ cannot distinguish genuine distribution shift from sampling noise, a limitation that does not invalidate the theorem (whose shift term uses the population-level $W_{1,d}(\mu, \nu) = 0$ in this setting) but that users of the diagnostic framework should keep in mind.

**Assumption S4 diagnostics.** Because hidden-state Sobolev propagation is the strongest assumption in the proof, we report layer-wise $H^s$ statistics in Figure 5. Across source/target distributions and train/test meshes, measured hidden-state norms remain finite and layer-structured for both constrained and unconstrained models. The constrained model exhibits a lower and more stable envelope, consistent with its smaller explicit constants and reduced bound-to-excess gap in Experiment 1. These measurements do not prove S4, but they provide direct empirical support that the assumption tracks realized hidden-state behavior in the studied regimes.

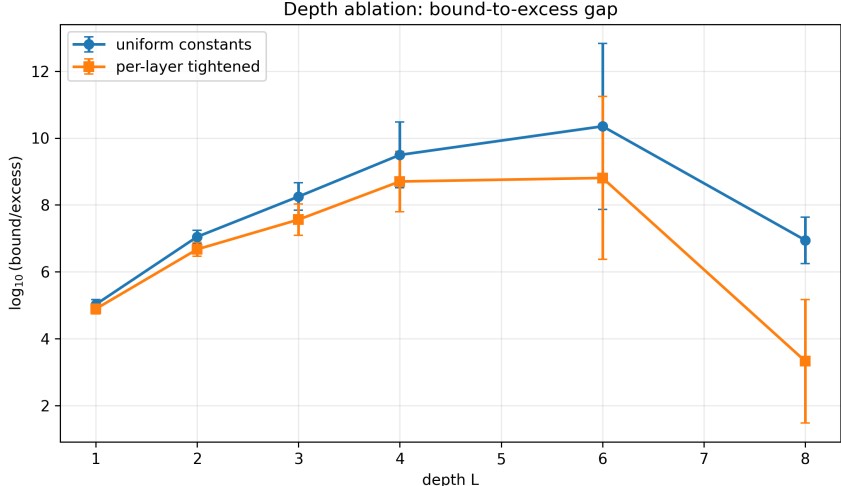

Figure 4: Depth ablation: $\log_{10}(\text{bound}/\text{excess})$ versus depth with 10 seeds per depth. Per-layer-tightened constants (lower curve) consistently improve over the uniform envelope.

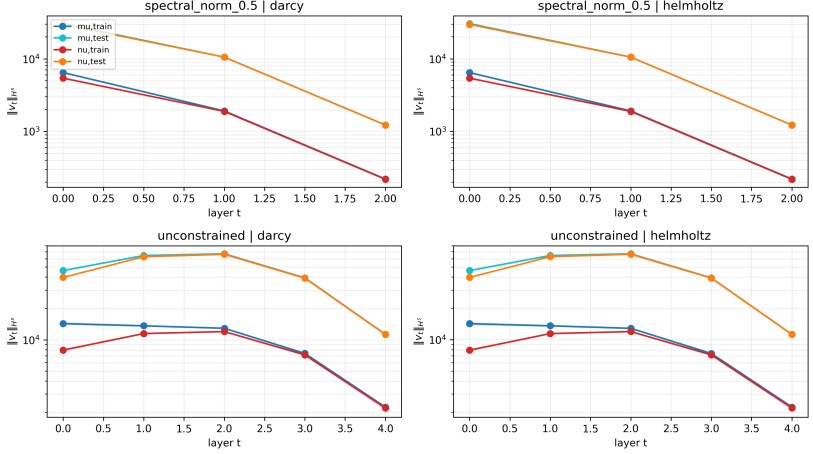

Figure 5: Layer-wise empirical $H^s$ hidden-state norms across source/target distributions and train/test meshes. The spectrally constrained model has a tighter and more stable envelope than the unconstrained model.

## 8 LIMITATIONS AND SCOPE

The theorem applies to periodic domains, smooth activations, and uniform grids. Non-periodic boundaries, irregular meshes, and non-smooth activations are outside the proved setting.

The bound is conservative in the tested regimes, with empirical bound-to-excess gaps of six to nine orders of magnitude. We therefore interpret the result as a decomposition and scaling statement rather than a tight predictor of test error.

The global Lipschitz envelope can grow exponentially with depth in worst case, which is the primary source of bound looseness. Local directional Lipschitz diagnostics (computed but not tabulated here) suggest the realized sensitivity is far below the worst-case envelope. These diagnostics are not part of the theorem.

Hidden-state Sobolev propagation (S4) is derived for $s=2$, $d=2$ in Proposition 1 (Appendix F). The general fractional-$s$ case remains an explicit assumption.

The aliasing envelope scales as $(2K + 1)N^{-s}$ (Appendix E). If $K$ grows proportionally with $N$, this becomes $N^{1-s}$. Hence the attainable rates deteriorate as mode budgets approach Nyquist, and interpretation should focus on moderate $K/N$ regimes.

**Open problems.** Extending to non-periodic domains requires replacing the Fourier aliasing analysis with analogous results for other spectral bases (wavelets, spherical harmonics). The four-channel decomposition structure would be preserved. Replacing the global Lipschitz envelope with local or average-case estimates could tighten the bound by orders of magnitude. Our directional Lipschitz diagnostics show realized sensitivity 5–7 orders below the worst-case envelope. Corollary 2 demonstrates modular replacement of the complexity channel via PAC-Bayes. Extending similar improvements to the discretization, shift, and aliasing channels is a natural next step. Finally, the additive structure absorbs cross-channel interactions into envelope constants. Developing multiplicative or mixed decompositions could further reduce conservatism.

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

## A  EXTENDED EMPIRICAL EVIDENCE

To complement the targeted diagnostics above, we include sweep-level summaries and one fixed-sample mesh-transfer visualization. Sweep-level aggregates (Table 5) confirm persistent bound conservatism across regimes. The per-channel breakdown is available in Table 5, which reports complexity, discretization, shift, aliasing, and concentration contributions separately. The shift column is nearly constant across model sizes within each PDE ($\approx 3.15$ for Darcy, $\approx 3.11$ for Helmholtz) because the projected $\widetilde{W}_1$ proxy depends on the distribution pair, not the model.

## B  CONSTANT-LEVEL DERIVATIONS

This appendix provides concrete prefactors for the terms in Theorem 1.

Table 5: Comprehensive Darcy/Helmholtz sweep summary by PDE and model size. We report the nonnegative seed-best excess proxy and means of all theorem addends.

| pde | size | n_runs | bound_rhs | excess_proxy | complexity | discretization | shift | aliasing | concentration | $\log_{10}(\text{bound/excess})$ |
|---|---|---|---|---|---|---|---|---|---|---|
| darcy | medium | 200 | 1.158e+05 | 8.053e-05 | 1.158e+05 | 5.280e-04 | 3.151e+00 | 4.407e-04 | 1.306e-01 | 9.16 |
| darcy | small | 200 | 1.498e+03 | 4.521e-05 | 1.495e+03 | 1.032e-03 | 3.151e+00 | 7.630e-04 | 1.306e-01 | 7.52 |
| helmholtz | medium | 200 | 1.642e+01 | 2.628e-07 | 1.319e+01 | 2.709e-06 | 3.108e+00 | 2.310e-06 | 1.306e-01 | 7.80 |
| helmholtz | small | 200 | 1.140e+02 | 1.142e-06 | 1.108e+02 | 4.641e-05 | 3.108e+00 | 4.582e-05 | 1.306e-01 | 8.00 |

Table 6: Additional isolated Burgers-1D evidence under viscosity shift.

| PDE | Bound RHS | Excess (seed-best) | Discretization | Shift |
|---|---|---|---|---|
| burgers1d | 6.475e+00 | 3.179e-02 | 1.552e-03 | 5.020e-01 |

## A. GLOBAL LIPSCHITZ ENVELOPE FOR FNO

The main difficulty in establishing a global Lipschitz bound is that the product of per-layer constants grows exponentially in depth. We now derive this bound explicitly.

Consider one hidden layer

$$\Phi_t(v) = \sigma(W_t v + \mathcal{K}_t v + b_t).$$

Assume

$$\|W_t\|_{L^2 \to L^2} \le M_\theta, \qquad \|\mathcal{K}_t\|_{L^2 \to L^2} \le M_\theta, \qquad \|\sigma'\|_\infty \le B_\sigma.$$

Then for any $v, \tilde{v}$,

$$\|\Phi_t(v) - \Phi_t(\tilde{v})\|_{L^2} \le B_\sigma \|W_t(v - \tilde{v}) + \mathcal{K}_t(v - \tilde{v})\|_{L^2} \le 2B_\sigma M_\theta \|v - \tilde{v}\|_{L^2}.$$

Iterating across depth-$L$ hidden stack:

$$\|\Phi_{1:L}(v) - \Phi_{1:L}(\tilde{v})\|_{L^2} \le (2B_\sigma M_\theta)^L \|v - \tilde{v}\|_{L^2}.$$

Including input/output projections $(\mathcal{P}_{\text{in}}, \mathcal{P}_{\text{out},1}, \mathcal{P}_{\text{out},2})$ gives the admissible bound

$$L_G \le \|\mathcal{P}_{\text{out},2}\| \cdot B_\sigma \|\mathcal{P}_{\text{out},1}\| \cdot (2B_\sigma M_\theta)^L \cdot \|\mathcal{P}_{\text{in}}\|.$$

This worst-case envelope scales exponentially in depth and is often numerically vacuous in practice. We use it only to establish admissible constants.

## B. COMPLEXITY TERM CONSTANT

Let $\ell$ be $L_\ell$-Lipschitz and $|\ell| \le R_\ell$ on the bounded regime. Then

$$\sup_{G \in \mathcal{G}} \left| \widehat{\mathcal{E}}_{\mu,h}(G) - \mathcal{E}_{\mu,h}(G) \right| \le 2L_\ell \mathfrak{R}_n(\mathcal{G}) + 2R_\ell \sqrt{\frac{\log(1/\delta)}{n}}.$$

Proof ingredients are standard: symmetrization,

$$\mathbb{E} \sup_G |\widehat{\mathcal{E}} - \mathcal{E}| \le 2\mathfrak{R}_n(\ell \circ \mathcal{G}),$$

contraction

$$\mathfrak{R}_n(\ell \circ \mathcal{G}) \le L_\ell \mathfrak{R}_n(\mathcal{G}),$$

and bounded-difference concentration.

**Remark 4** (Explicit complexity rate). *Under Assumption 1, the covering-number argument of Kim and Kang Kim & Kang (2024) gives $\mathfrak{R}_n(\mathcal{G}_{K,L,W}; h) \le 2L_G B_x/\sqrt{n}$, where $L_G$ is the global Lipschitz envelope (Section A) and $B_x$ the input bound from (S1). Substituting yields $C_1 \mathfrak{R}_n \le 8L_\ell L_G B_x/\sqrt{n}$, confirming the $n^{-1/2}$ statistical scaling.*

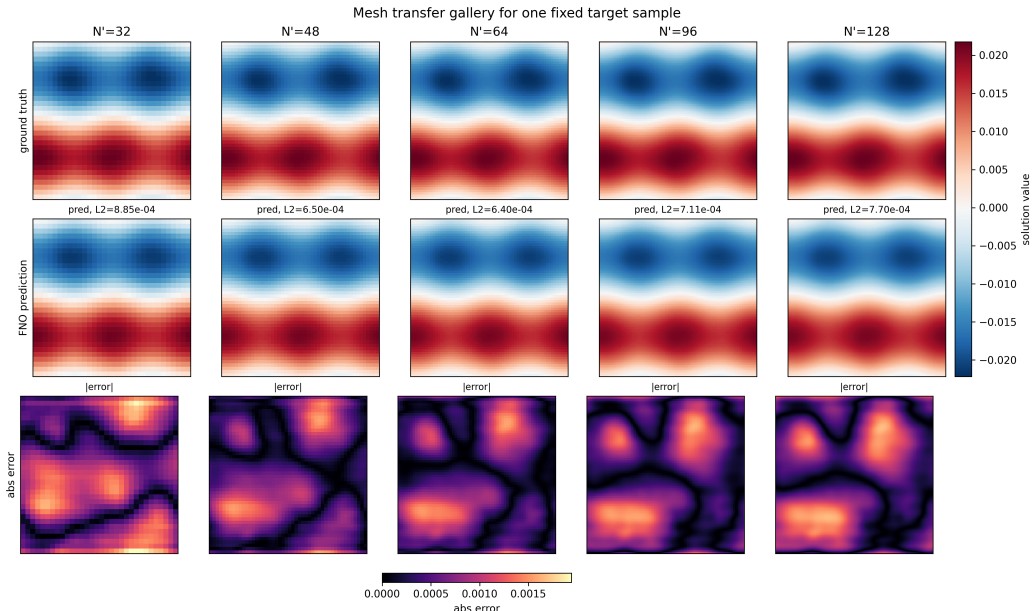

Figure 6: Mesh-transfer gallery for one fixed target sample across $N' \in \{32, 48, 64, 96, 128\}$ with shared color scales.

## C. DISCRETIZATION TERM CONSTANT

For $f \in H^s(\mathbb{T}^2)$ Adams & Fournier (2003); Evans (2010), $s > 1$, trigonometric interpolation yields

$$\|f - \mathcal{I}_h f\|_{L^2} \leq C_{\mathrm{Sob}} h^s \|f\|_{H^s}.$$

Let $v_t$ be continuum hidden state and $v_t^h$ its grid-$h$ counterpart with shared parameters. Define $e_t = \|v_t - v_t^h\|_{L^2}$. One step gives

$$e_{t+1} \leq 2B_\sigma M_\theta e_t + \eta_t,$$

where interpolation defect $\eta_t$ satisfies

$$\eta_t \leq C_{\mathrm{Sob}} h^s \|v_t\|_{H^s}.$$

Under Assumption 1(S4), i.e., bounded hidden-state Sobolev norm $\|v_t\|_{H^s} \leq C_{\mathrm{hid}}$:

$$e_{t+1} \leq 2B_\sigma M_\theta e_t + C_{\mathrm{Sob}} C_{\mathrm{hid}} h^s.$$

Unrolling the recursion yields

$$e_L \leq C_{\mathrm{Sob}} C_{\mathrm{hid}} h^s \sum_{j=0}^{L-1} (2B_\sigma M_\theta)^j.$$

Hence

$$T_2 \leq L_\ell C_{\mathrm{disc}} h^s, \qquad C_{\mathrm{disc}} = C_{\mathrm{Sob}} C_{\mathrm{hid}} \sum_{j=0}^{L-1} (2B_\sigma M_\theta)^j.$$

## D. SHIFT TRANSFER CONSTANT

For $\varphi_G(a, u) = \ell(G_h(a), u)$ on $(\mathcal{X} \times \mathcal{Y}, d)$, if $\varphi_G$ is $L_\ell(1 + L_G)$-Lipschitz, Kantorovich–Rubinstein implies

$$|\mathbb{E}_\nu \varphi_G - \mathbb{E}_\mu \varphi_G| \leq L_\ell(1 + L_G) W_{1,d}(\mu, \nu).$$

Supremum over $G \in \mathcal{G}$ preserves the same prefactor. The theorem is stated with Wasserstein distance on the full sample space $\mathcal{X} \times \mathcal{Y}$. Experiments use a projected surrogate $\widetilde{W}_1$ on finite-dimensional embeddings for computability.

## E. ALIASING MISMATCH VIA LATTICE COLLISION DECOMPOSITION

The key mechanism is Fourier aliasing. Sampling on a finite grid folds high-frequency content into low modes, and different grids fold differently. We now quantify this mismatch for FNO hidden states across mismatched meshes.

Write Fourier expansion

$$f(x) = \sum_{k \in \mathbb{Z}^2} \widehat{f}(k) e^{2\pi i k \cdot x}.$$

The low/high-mode folding mechanism is the standard Fourier aliasing effect from spectral methods Trefethen (2000); Canuto et al. (2006). Let $D_N$ denote periodic sampling on $N \times N$ grid and $D_N^{-1}$ the trigonometric reconstruction map. For retained mode $\|k\|_\infty \le K$ ($K < N/2$),

$$(\widehat{D_N^{-1} D_N f})(k) = \sum_{m \in \mathbb{Z}^2} \widehat{f}(k + mN).$$

Hence low-band alias contamination

$$\varepsilon_{N,K}(f) := \Pi_{\le K}(D_N^{-1} D_N f - f)$$

has coefficients

$$\widehat{\varepsilon_{N,K}}(k) = \sum_{m \in \mathbb{Z}^2 \setminus \{0\}} \widehat{f}(k + mN), \qquad \|k\|_\infty \le K.$$

For $s > 1$, Cauchy–Schwarz with Sobolev weights gives

$$\sum_{m \ne 0} \left| \widehat{f}(k + mN) \right| \le \left( \sum_{m \ne 0} (1 + \|k + mN\|_2^2)^{-s} \right)^{1/2} \|f\|_{H^s} \le C_s N^{-s} \|f\|_{H^s},$$

because $K < N/2$ implies $\|k + mN\|_2 \ge |m|N - K \ge |m|N/2$, hence

$$\sum_{m \ne 0} (1 + \|k + mN\|_2^2)^{-s} \le C \sum_{m \ne 0} (|m|N)^{-2s} \le C_s^2 N^{-2s},$$

and $s > 1$ guarantees summability in two dimensions. Summing over $(2K + 1)^2$ retained modes:

$$\|\varepsilon_{N,K}(f)\|_{L^2} \le C_s(2K + 1)N^{-s}\|f\|_{H^s}.$$

Applying the single-grid estimate to $N$ and to $N'$, then using triangle inequality:

$$\|\Pi_{\le K}(D_N^{-1} D_N f - D_{N'}^{-1} D_{N'} f)\|_{L^2} \le C_{s,K}(N^{-s} + N'^{-s})\|f\|_{H^s},$$

with $C_{s,K} = C_s(2K + 1)$. The aliasing argument requires strict Nyquist separation $K < \min(N, N')/2$.

Now define hidden-state mismatch

$$\Delta_t := \Pi_{\le K} \mathcal{I}_{h' \leftarrow h} v_t^{(N)} - \Pi_{\le K} v_t^{(N')}.$$

Assume the input lifting is mesh-consistent in the retained band so that $\Delta_0 = 0$. Each layer satisfies

$$\|\Delta_{t+1}\|_{L^2} \le 2B_\sigma M_\theta \|\Delta_t\|_{L^2} + \beta_{s,K}(h^s + h'^s),$$

where $\beta_{s,K} = C_{s,K} C_{\mathrm{hid}}$ uses uniform hidden-state $H^s$ bound. With $\Delta_0 = 0$, recursion yields

$$\|\Delta_L\|_{L^2} \le \beta_{s,K}(h^s + h'^s) \sum_{j=0}^{L-1} (2B_\sigma M_\theta)^j.$$

Therefore

$$\mathcal{A}_{\sup}(K, h, h') \le C_{\mathrm{alias}}(L, M_\theta, B_\sigma, K, s)(h^s + h'^s),$$

with admissible choice

$$C_{\mathrm{alias}} = \beta_{s,K} \sum_{j=0}^{L-1} (2B_\sigma M_\theta)^j.$$

Since loss is $L_\ell$-Lipschitz,

$$T_4 \le L_\ell \, \mathcal{A}_{\sup}(K, h, h').$$

Also, by construction,

$$\overline{\mathcal{A}}_{\mu,\nu}(K, h, h') \le 2\mathcal{A}_{\sup}(K, h, h').$$

**Resulting constants.**  Sections A–E supply concrete prefactors for every term in Theorem 1 under assumptions (A1)–(A5). Together they complete the proof of the stated decomposition in the periodic Sobolev setting.

F. SUFFICIENT CONDITIONS FOR HIDDEN-STATE SOBOLEV PROPAGATION (S4)

This section derives Assumption (S4) from first principles for the case $s = 2$, $d = 2$ (integer Sobolev exponent on $\mathbb{T}^2$), replacing the stated assumption with an explicit proposition.

**Proposition 1** (Hidden-state Sobolev propagation). *Let $s = 2$, $d = 2$. Suppose the activation $\sigma \in C^2(\mathbb{R})$ satisfies $\|\sigma'\|_\infty \leq B_\sigma$ and $\|\sigma''\|_\infty \leq B'_\sigma$, and the FNO class satisfies (S1)–(S3). Then for every $G \in \mathcal{G}_{K,L,W}$ and every input $a$ with $\|a\|_{H^s} \leq B_x$, the hidden states satisfy $\|v_t\|_{H^s} \leq R_t$ for $t = 0, \dots, L$, where $R_t$ is defined by the recursion*

$$R_0 = \|\mathcal{P}_{\text{in}}\| \cdot B_x + C_{\text{grid}}, \tag{2}$$

$$R_{t+1} = \alpha_t R_t + \beta R_t^2, \tag{3}$$

*with per-layer linear amplification $\alpha_t = B_\sigma(\|\mathcal{K}_t\|_{L^2 \to L^2} + \|W_t\|_{L^2 \to L^2})$, quadratic coefficient $\beta = B'_\sigma C_{\text{GN}}^2 \max_t(\|\mathcal{K}_t\| + \|W_t\|)^2$, Ladyzhenskaya constant $C_{\text{GN}} = \sqrt{2}$ on $\mathbb{T}^2$, and $C_{\text{grid}}$ accounting for appended grid coordinates. Hence $C_{\text{hid}} = \max_{0 \leq t \leq L} R_t$ is an admissible choice for (S4).*

*Proof.* The proof proceeds by induction on $t$.

*Step 1: Linear operators preserve $H^s$ norms.* The spectral convolution $\mathcal{K}_t$ acts on retained modes $\|k\|_\infty \leq K$ via a per-mode channel-mixing matrix $\mathbf{M}_t(k) \in \mathbb{C}^{W \times W}$ with $\|\mathbf{M}_t(k)\|_2 \leq M_\theta$. Since Sobolev weights factor per-mode:

$$\|\mathcal{K}_t(v)\|_{H^s}^2 = \sum_{\|k\| \leq K} (1 + |k|^2)^s |\mathbf{M}_t(k)\hat{v}(k)|^2 \leq M_\theta^2 \sum_{\|k\| \leq K} (1 + |k|^2)^s |\hat{v}(k)|^2 \leq M_\theta^2 \|v\|_{H^s}^2.$$

Similarly, the pointwise convolution $W_t$ (1×1 in space, constant across frequencies) satisfies $\|W_t(v)\|_{H^s} \leq \|W_t\|_2 \|v\|_{H^s} \leq M_\theta \|v\|_{H^s}$. Therefore:

$$\|\mathcal{K}_t(v) + W_t(v)\|_{H^s} \leq (\|\mathcal{K}_t\| + \|W_t\|)\|v\|_{H^s}.$$

*Step 2: Sobolev chain rule for activation.* For $s = 2$ (integer), $f \in H^2(\mathbb{T}^2; \mathbb{R}^W)$, and $\sigma \in C^2$, the standard chain rule gives:

$$\nabla\sigma(f) = \sigma'(f)\nabla f,$$
$$\nabla^2\sigma(f) = \sigma''(f)(\nabla f)^{\otimes 2} + \sigma'(f)\nabla^2 f.$$

Therefore:

$$\|\sigma(f)\|_{L^2} \leq B_\sigma\|f\|_{L^2},$$
$$\|\nabla\sigma(f)\|_{L^2} \leq B_\sigma\|\nabla f\|_{L^2},$$
$$\|\nabla^2\sigma(f)\|_{L^2} \leq B'_\sigma\|\nabla f\|_{L^4}^2 + B_\sigma\|\nabla^2 f\|_{L^2}.$$

By the periodic Ladyzhenskaya inequality on $\mathbb{T}^2$, $\|\nabla f\|_{L^4} \leq C_{\text{GN}}\|f\|_{H^2}$ with $C_{\text{GN}} = \sqrt{2}$. Combining:

$$\|\sigma(f)\|_{H^2} \leq B_\sigma\|f\|_{H^2} + B'_\sigma C_{\text{GN}}^2\|f\|_{H^2}^2.$$

*Step 3: One-step recursion.* At layer $t$, the pre-activation state is $z_t = \mathcal{K}_t(v_t) + W_t(v_t)$. Assuming $\|v_t\|_{H^s} \leq R_t$:

$$\|z_t\|_{H^s} \leq (\|\mathcal{K}_t\| + \|W_t\|)R_t.$$

Applying Step 2 to $v_{t+1} = \sigma(z_t)$:

$$\|v_{t+1}\|_{H^s} \leq B_\sigma(\|\mathcal{K}_t\| + \|W_t\|)R_t + B'_\sigma C_{\text{GN}}^2(\|\mathcal{K}_t\| + \|W_t\|)^2 R_t^2.$$

Since $(\|\mathcal{K}_t\| + \|W_t\|)^2 \leq \max_t(\|\mathcal{K}_t\| + \|W_t\|)^2$, this yields $\|v_{t+1}\|_{H^s} \leq \alpha_t R_t + \beta R_t^2 = R_{t+1}$.

The base case $\|v_0\|_{H^s} \leq \|\mathcal{P}_{\text{in}}\|\|a\|_{H^s} + C_{\text{grid}} \leq R_0$ follows from the same linear argument applied to the input projection plus bounded grid features. $\square$

**Remark 5** (Contractive regime). *When $\max_t \alpha_t < 1$ and $R_0 < (1 - \max_t \alpha_t)/\beta$, the recursion* (3) *converges and the base case* (2) *determines $C_{\mathrm{hid}} = R_0$. This regime is achievable via spectral norm constraints that enforce $B_\sigma(\|\mathcal{K}_t\| + \|W_t\|) < 1$ for all $t$.*

**Remark 6** (Extension to non-integer $s \in (1,2)$). *For fractional $s$, the explicit chain rule is replaced by the fractional Leibniz rule (Kato–Ponce inequality), which introduces an abstract constant $C_{\mathrm{frac}}(s,d)$ in place of $C_{\mathrm{GN}}^2$. The recursion structure is unchanged. Only the quadratic coefficient $\beta$ differs.*

## G. PAC-Bayes Complexity Variant (Corollary 2)

The proof of Theorem 1 decomposes excess risk into three groups ($\Gamma_{\mathrm{gen}}$, $\Gamma_{\mathrm{shift}}$, $\Gamma_{\mathrm{mesh}}$) via equation (1). Lemmas 2 and 3, which bound $\Gamma_{\mathrm{shift}}$ and $\Gamma_{\mathrm{mesh}}$, depend only on Lipschitz properties of the loss and the operator class. They are independent of the generalization mechanism used for $\Gamma_{\mathrm{gen}}$.

To obtain the PAC-Bayes variant, we replace Lemma 1 with the McAllester bound McAllester (1999). Let $P$ be a prior distribution on the parameter space chosen before seeing data, and let $Q$ be a (possibly data-dependent) posterior. The PAC-Bayes theorem gives, with probability at least $1 - \delta$ over the training sample,

$$\mathbb{E}_{G \sim Q}[\mathcal{E}_{\mu,h}(G)] - \mathbb{E}_{G \sim Q}\left[\widehat{\mathcal{E}}_{\mu,h}(G)\right] \leq \sqrt{\frac{\mathrm{KL}(Q\|P) + \log(2\sqrt{n}/\delta)}{2n}}.$$

For Gaussian prior $P = \mathcal{N}(w_0, \lambda^2 I)$ and posterior $Q = \mathcal{N}(\hat{w}, \lambda^2 I)$, the KL divergence is $\|\hat{w} - w_0\|^2/(2\lambda^2)$.

Substituting this bound for $\Gamma_{\mathrm{gen}}$ into the decomposition and retaining Lemmas 2 and 3 unchanged yields Corollary 2. The parameter $\lambda$ can be optimized post hoc to minimize the bound; this does not affect the validity of the high-probability guarantee.

This modularity illustrates a broader point. The four-channel structure separates the generalization mechanism from the mesh-transfer and distribution-shift analyses. Any improved bound on $\Gamma_{\mathrm{gen}}$ (PAC-Bayes, compression-based, or margin-based) can be substituted without rederiving the other three channels.

