# OpenReview forum: "Four-Term Finite-Sample OOD Transfer Bound for Fourier Neural Operators"
_mathai.club/MathAI/2026/Conference — 2026 Oral_

### Official Review · Reviewer_yAsf · 2026-03-12
**Accurate theoretical unification of FNO degradation risks.**

**Rating:** 7
**Confidence:** 4

**Review:**

Authors derive finite-sample excess-risk framework for nonlinear FNOs under distribution shift + mesh transfer, decomposing 4 degradation channels.
Pros:
First unified theorem combining complexity + discretization + Wasserstein shift + mesh-mismatch aliasing
Complete proof with explicit constants in Sobolev spaces
Experiments confirm predicted scaling trends on Darcy/Helmholtz/Burgers
Cons:
Bound conservative (gap ~10⁶–10⁸ vs. empirical excess risk)
Limited to periodic domains, smooth activations, uniform grids
Lipschitz constant grows exponentially with depth
Conclusion:
For engineers working with FNO within 4 depth layers density results of this work could be highly applicable.

---

### Decision · Program_Chairs · 2026-03-14

**Decision:**

Accept (Oral)

**Comment:**

Dear Author(s),

On behalf of the Program Committee of the International Conference on Mathematics of Artificial Intelligence (MathAI 2026), we are pleased to inform you that your paper has been accepted for an oral presentation at MathAI 2026.

Your paper was evaluated through a rigorous two-stage review process involving both automated screening and expert review by members of the Program Committee. The reviewers recognized the quality and contribution of your work.

Presentation details:

- Format: Oral presentation (15–20 minutes + 5 minutes Q&A)
- Mode: You may present either in person (offline) at the conference venue in Sirius, Russia, or remotely via Zoom. Please indicate your preferred mode when confirming your participation.
- Conference dates: Marh 30 - April 3, 2026
- Website: https://mathai.club

Next steps:

1. Please confirm your participation and presentation mode by replying to this email mathai.club@yandex.ru no later than March 15, 2026 18:00 Moscow time.
2. If you plan to attend in person, the organizing committee will provide accommodation details separately.
3. Please prepare your final camera-ready manuscript according to the formatting guidelines available at https://mathai.club and upload it to OpenReview by March 15, 2026 18:00 Moscow time.

Should you have any questions regarding the program, logistics, or your presentation slot, please do not hesitate to contact us.

We look forward to your contribution to MathAI 2026.

With kind regards,

MathAI 2026 Program Committee
International Conference on Mathematics of Artificial Intelligence
https://mathai.club
OpenReview: https://openreview.net/group?id=mathai.club/MathAI/2026/Conference
Telegram: https://t.me/MathAI_club
Email: mathai.club@yandex.ru